# Avian Scavengers as Bioindicators of Antibiotic Resistance Due to Livestock Farming Intensification

**DOI:** 10.3390/ijerph17103620

**Published:** 2020-05-21

**Authors:** Guillermo Blanco, Luis M. Bautista

**Affiliations:** Department of Evolutionary Ecology, Museo Nacional de Ciencias Naturales (MNCN-CSIC), José Gutiérrez Abascal 2, 28006 Madrid, Spain; lm.bautista@csic.es

**Keywords:** antibiotics, bacterial resistance, birds, carcass dumps, *Escherichia coli*, livestock intensification, *Milvus*

## Abstract

Industrial food animal production uses huge amounts of antibiotics worldwide. Livestock, their excreta used for manure and meat subproducts not intended for human consumption can all play important roles in the transmission of bacterial resistance to wildlife. Vultures and other scavengers can be directly exposed to active antibiotics ingested while feeding on livestock carcasses. This study evaluates whether bacterial resistance in the red kite (*Milvus milvus*) differs between two wintering areas selected based on patent differences in farming practices—particularly in the industrial production of food animals (primarily swine and poultry) vs. scarce and declining sheep herding. The results support the hypothesis that intensification in food animal production is associated with increased bacterial multidrug resistance in wildlife. Resistance was positively correlated with time elapsed since the beginning of the commercial application of each antibiotic in human and veterinary medicine, with clear differences depending on farming intensification between areas. Monitoring programs are encouraged to use red kites and other avian scavengers as valuable sentinels of contamination by antibiotics and clinically relevant resistant pathogens from livestock operations of variable intensities. Farms authorized for supplementary feeding of threatened scavengers should avoid supplying carcasses with active antibiotic residues to avoid bacterial resistance in scavenger wildlife.

## 1. Introduction

The intensification of livestock production and management systems has spawned the use of antibiotics and other pharmaceuticals to mitigate disease worldwide [1]. These practices have been directly associated with increased levels of bacterial resistance in humans, livestock and the environment, to the point that animal husbandry facilities have become antibiotic-resistance hotspots [2,3,4,5]. High loads of microbes and antibiotics used to combat disease contribute to the release of resistance agents in farmland environments, especially from bacteria and drugs discharged through manuring in agricultural land [6,7]. In addition, the presence of potentially pathogenic bacteria from livestock and humans coupled with insufficient hygiene and drug residues in carcass and rubbish dumps [8,9] can increase antibiotic bacterial resistance in wildlife frequenting these places as foraging grounds [10,11]. Bacterial resistance can be host-amplified, harbored and spread by wildlife in humanized landscapes, representing a concerning, but not fully understood, factor in the One Health framework [12].

In contrast with most wildlife species, which are generally only exposed to resistant bacteria from human activities polluting the environment [13,14], obligate and facultative scavengers like vultures and eagles can also be directly exposed to active antibiotics ingested while feeding on livestock carcasses [15,16,17]. Withdrawal times to prevent harmful drug residues in the meat that humans consume do not apply to food animal byproducts not intended for human consumption, which are often disposed in carcass dumps for avian scavengers [8]. These places represent sources of abundant and predictable food concentrating large numbers of these birds, which can increase acquisition, intra- and interspecific cross-transmission and spread of livestock, wildlife and human pathogens [9,11]. Carrion from medicated livestock can thus pose threats to wildlife health due to infection risk coupled with pollution by harmful drugs and their resistance determinants. This concerning combination of factors can further increase the resistance or cross-resistance of pathogenic and commensal bacteria in wildlife. In addition, the selective pressure of antibiotics ingested while feeding on medicated livestock carcasses can promote novel bacteria–antibiotic resistance interactions in natural environments [18,19], especially in host-specific or dominant wildlife microbiota [20,21,22]. This can be triggered by the regular variable-dose exposure to multiple antibiotics differentially administered in the suite of farming environments and livestock species exploited by these mobile species [23,24], similarly to the misuse of these drugs in the emergence of bacterial resistance in human and livestock populations [25,26,27].

The history, from the first hints of penicillin resistance to the increasingly rare last-resort antibiotics, is well known in human and veterinary medicine [25,28]. Wildlife whose diet includes meat from feeding on livestock carcasses can show a comparable resistance trend in their gut microbiome, but evidence is lacking for these species in the wild. Therefore, surveillance of resistant bacteria in wildlife is warranted to fully understand the patterns of emergence, acquisition and spread of resistance determinants across human–livestock–wildlife interfaces [5]. Wildlife exposure to antibiotics, other pharmaceuticals and drug cocktails that can cause mortality and disease in avian scavengers [16,29,30,31] make these species useful bioindicators of microbial and pharma-pollution in the environment, with implications in wildlife and public health. However, these species have not been, to our knowledge, used in pharmacovigilance programs or antibiotic-resistance risk assessment in farmlands and other humanized landscapes.

In this study, we evaluated whether contrasting intensification in livestock management influences antibiotic resistance in an avian scavenger, the red kite (*Milvus milvus*), long associated with farmlands, including the exploitation of abundant, concentrated and predictable food at carcass dumps [32]. Resistance patterns to 20 antibiotics determined in gut bacteria (*Escherichia coli*) from red kite droppings were compared between two areas selected based on patent differences in farming practices, particularly in the industrial production of food animals (primarily swine and poultry) vs. scarce and declining sheep herds [32]. Previous studies showed that the more intensive the production of food animals, the heavier the active antibiotic residues and the resistance agents discharged into the environment [5,33]. Therefore, we predict a greater antibiotic bacterial resistance in red kites from the area with factory farms compared to those from areas with traditional extensive shepherding. Finally, we evaluated whether the relationship between increasing bacterial resistance and the time elapsed since the beginning of the commercial application of each antibiotic used in human and veterinary medicine was associated with the resistance patterns in wildlife depending on livestock intensification. This aimed to assess the value of avian scavengers as bioindicators of contamination with antibiotics and resistant bacteria from livestock operations.

## 2. Materials and Methods

### 2.1. Study Species and Areas

The red kite is a medium-sized (~1 kg) facultative scavenger endemic to Europe. Individuals from northern latitudes are migrant birds wintering in the Iberian Peninsula, while individuals breeding in Iberia are year-round residents. This species has suffered a sharp decline within their breeding and wintering ranges during the last decades. Human persecution through illegal and unintended poisoning, alteration of their food sources, contamination, genetic erosion, and the combined effect on reduced breeding success have been highlighted as the main causes of decline [34,35,36,37].

The study was conducted in two areas of central Spain (Madrid and Segovia) selected based on their contrasting low and high intensification of livestock, respectively. The northeast area of Madrid province is a plain, mostly devoted to dry cereal crops, where livestock farming is reduced to small sheep numbers grazing under extensive conditions. No carcass dumps exist in this area. The application of manure from livestock in this area is limited to the dung from sheep grazing freely. Red kites mostly forage solitarily, dispersed across the countryside, on wild prey and use a single roosting site. In contrast, the central area of Segovia province supports large numbers of stabled livestock (especially swine and poultry) reared under industrial conditions including intensive medication with multiple pharmaceuticals. Carcasses of these livestock have long been used to feed avian scavengers and their manure spread as fertilizer in agricultural lands [38]. Segovia area has traditionally been used by large numbers of wintering red kites (about 2300 individuals in the study period) gathering in multiple communal roosts and exploiting livestock carcasses dumped near farms and in supplementary feeding stations intended for avian scavenger conservation. Details on the potential impacts of feeding on carcasses of intensive livestock carcasses in supplementary feeding stations on red kites health are provided in previous studies, including overcrowding, diet simplification and infection with internal parasites and pathogens [32,36].

### 2.2. Fieldwork

In February 2013, 36 and 48 fresh feces were collected beneath the trees exclusively used by communally roosting red kites in the areas with low-intensity (Madrid) and high-intensity (Segovia) farming, respectively (see details in [32]). Fresh feces were sampled with sterile microbiologic swabs and Amies transport medium, no more than 1 h after their deposition, in an attempt to avoid desiccation of the feces produced at dawn. The kites typically distributed homogeneously through the roosting trees, which allowed the sampling of non-adjacent feces (separated by more than 5 m) to avoid sample duplication from the same individuals. Samples were transported in a container with ice to the laboratory (Laboratorio Regional de Sanidad Animal, Consejería de Medio Ambiente y Ordenación del Territorio, Comunidad de Madrid, Colmenar Viejo, Madrid, Spain) on the same day of collection and processed within one to two hours following their arrival.

### 2.3. Microflora Culture and Antibiotic Resistance Identification

Samples were cultured with standard methods. Briefly, samples were cultured in 5 % sheep blood and MacConkey agar in aerobic and anaerobic conditions, and plates were incubated at 37 °C, for 24 h. All suspect colonies were subcultured on appropriate medium and *E. coli* identified by using multi-substrate identification stripes (API 20E, Bio Merieux, Marcy l’Etoile, France) as described previously [39]. Susceptibility of *E. coli* to selected antibiotics commonly used in livestock farming and human medicine was tested with the Kirby–Bauer disk diffusion method, which was performed and interpreted as described by the Clinical and Laboratory Standards Institute CLSI protocols [40]. Well-isolated bacterial colonies were selected from an agar plate culture and transferred into a broth culture, which was incubated at 37 °C until a slight visible turbidity appeared (similar to 0.5 McFarland standard), usually within 2 to 6 h. A sterile swab was dipped into the standardized suspension of bacteria and excess fluid was removed by pressing and rotating the swab firmly against the inside of the tube above the fluid level. The plates were incubated at 37 °C for 24 h and zones of inhibition were measured. Commercial antibiotic disks (BD BBLTM Sensi-DiscTM, Franklin Lakes, NJ, USA) were used for a total of 20 tested antibiotics, including aminoglycosides, β-lactams, polypeptides, quinolones, sulfonamides and tetracyclines (see Table 1). We used a panel of antibiotics commonly used to test for bacterial resistance, combining compounds used for long time and other more recently included in veterinary and human medicine to have a more broad view of the problem—and because the antimicrobials used in Spain may differ from those used in the breeding areas of red kites in central Europe. Because some resistances could be also acquired by red kites foraging in rubbish dumps of urban refuse, we included several antibiotics generally not used in livestock as control to assess main sources of bacterial resistance. Each bacterial isolate was classified as susceptible, intermediate or resistant, depending on the growth inhibition diameter. Disk contents and growth inhibition zones were in accordance with criteria set by the CLSI standard protocol for bacteria isolated from animals.

### 2.4. Statistical Analyzes

The occurrence of isolates resistant and susceptible to each antibiotic was compared between study areas (low and high intensification) using a Fisher’s exact test. Although most isolates showed clear susceptibility or resistance, a proportion showed intermediate resistance towards particular antibiotics, which may be indicative of processes of emergence or loss of resistance. Therefore, these isolates were considered as resistant in order to simplify the analyses. Two-tailed tests were calculated, despite a one-tailed null hypothesis: the greater the intensification of livestock farming, the stronger the predicted resistance to antibiotics. Relative risk (RR) and 95% confidence intervals (CI) were calculated. The resistance patterns between areas were also analyzed by antibiotic family or subfamily (β-lactams) because the resistance mechanisms are expected to be related between antibiotics with a similar molecular structure.

The number of antibiotic agents and families to which each isolate was resistant was compared between areas with low and high livestock intensity with the Mann–Whitney *U* test and its statistical significance was calculated with the approximation of the Z distribution.

The relationship between bacterial resistance and the time elapsed since the beginning of the commercial application of each antibiotic (hereafter time elapsed) was tested using a generalized linear model (GLM) with binomial error and logit identity. The antibiotic activity of each isolate (susceptible = 0, resistant = 1) was the response variable, while the time elapsed (covariate), the low- or high-intensity area (factor) and the interaction of time elapsed x area were included as explanatory variables. Time elapsed is a fixed feature of each antibiotic, so it was nested within antibiotics because the two variables do not interact. Statistical analyses were performed using SPSS software v. 25 (IBM Corp., Armonk, NY, USA) [41] and JMP 12.0 (SAS Institute Inc., Cary, NC, USA) [42].

## 3. Results

*E. coli* was isolated in most droppings from the low- (44 out of 48, 91.7%) and high-intensity (35 out of 36, 97.2%) areas. High rates ( >50%) of resistant isolates to one aminoglycoside (neomycin) and one cephalosporin (cephalothin) were found in both areas, while other aminoglycosides (streptomycin) and the three tetracyclines tested showed high resistances only in the high-intensity area (Table 1). No isolates resistant to the only carbapenem and polypeptide tested were detected in either study areas (Table 1). The resistance to the remaining antibiotics tested was generally higher in the area with high livestock intensification. Overall, resistance to eight antibiotics was significantly higher in the high- compared to the low-intensity area (Table 1).

The proportion of isolates that were resistant to each antibiotic family was higher in the area with high farming intensification (Figure 1a), especially for aminoglycosides (Fisher’s exact test, *p* = 0.019, RR = 1.213−82.472) and tetracyclines (*p* = 0.001, RR = 1.808−13.143), but not significant for sulfonamides (sulfamethoxazole–trimethoprim, Table 1). The resistance to quinolones was low and did not differ between areas (*p* = 0.454, RR = 0.086−2.598). The comparison for β-lactams did not reach statistical significance (*p* = 0.077, RR = 0.907−13.938) due to the effect of antibiotic subfamilies (Figure 1b), especially ureidopenicillins (piperacillin, see Table 1) and cephalosporins (*p* = 0.045, RR = 1.030−15.534) and less due to aminopenicillins (*p* = 0.066, RR = 1.015−6.379).

A proportion of isolates (4 out of 44, 9.1%) from the low-intensity area were susceptible to all antibiotics tested, while all isolates from the high-intensity area showed resistance to at least one antibiotic (100%, *n* = 35). Multi-resistance was lower in the low-intensity area than in the high-intensity area (Figure 2a), both considering the number of agents (Z = 3.76, *n*_low_= 44, *n*_high_= 35, *p* <0.0001) and the number of antibiotic families (Z = 3.25 *n*_low_ = 44, *n*_high_= 35, *p* = 0.001). Overall, the distribution of multi-resistance to different antibiotic families differed between areas (χ^2^ = 14.50, *df* = 5, *p* = 0.013); isolates from the low intensity area were more frequently resistant to two antibiotic families, while those from the high-intensity area showed a similarly higher frequency of resistance to three and four families (Figure 2b).

About twice as many different patterns of resistance to single or multiple antibiotic families were recorded in the area with low and high livestock intensity, respectively (Table 2). The most frequent combinations were those involving resistance to aminoglycosides and β-lactams (A-B) in the low-intensity area, and those involving triple resistance to aminoglycosides, β-lactams and tetracyclines (A-B-T) and quadruple resistance to the same families plus sulfonamides (A-B-S-T) in the high-intensity area (Table 2).

The GLM testing the variation in bacterial resistance with the timeline of each antibiotic therapy showed higher values in the high- compared to the low-intensity area (χ^2^= 4.634, *df* = 1, *p* = 0.031), an increasing resistance with the time period elapsed (χ^2^ = 134.979, *df* = 18, *p* < 0.0001) and a significant interaction between time period elapsed and study area (χ^2^= 15.395, *df* = 1, *p* < 0.0001) indicating a more acute loss of susceptibility with time in the high- than the low-intensity area (Figure 3).

## 4. Discussion

Several studies have highlighted antibiotic resistance in wildlife owing to environmental spread of resistant strains from prevailing anthropogenic activities, including manuring, rubbish dumps, urbanization, wastewater facilities and sewage sludge systems [13,14]. Few studies have attempted to relate bacterial resistance to the food sources of wildlife considering the intensity of livestock farming and the intentional elimination of carcass residues for avian scavenger consumption [11]. The results of this study support the hypothesis that intensification in food animal production is associated with increased bacterial resistance to antibiotics in a facultative avian scavenger.

As predicted, higher bacterial resistance was found in red kites feeding on swine carcasses from intensive factory farms compared to those feeding on wild animals in farmlands with low-intensity sheep herding (see details on diet in [32,36]). These differences were especially high for tetracyclines, the antibiotics most intensively used in livestock farming over the last few decades in Spain [43]. Other antibiotic classes showing high resistance levels were also intensively used in livestock operations in Spain, especially aminoglycosides and β-lactams, although this may depend on particular agents and their administration form in each food-producing animal species [43]. Among β-lactams, resistance was especially high for the 4^th^ generation ureidopenicillin tested (piperacillin) and for the 1^st^ generation cephalosporins (cefalothin, cephalexin). The low or null resistance and the lack of differences between areas for 3^rd^ generation cephalosporins (ceftazidime), aminopenicillins (both 1^st^ and 3^rd^ generation), polypeptides, quinolones and sulfonamides suggest a recent or less generalized use in livestock operations [43]. These low resistances can be partially due to the chemical properties of each antibiotic, including their different pharmacokinetics and degradation. Some of these agents are considered as a last option in human and veterinary therapy, which makes them valuable tests in antibiotic-resistance risk assessment in wildlife as a consequence of future farming developments. Overall, multi-resistance was lower in the low- compared to the high-intensity area, for the number of both agents and families of antibiotics. These differences determined a more frequent combination of resistance to aminoglycosides and β-lactams in the low-intensity area, while resistances to these families plus sulfonamides and tetracyclines were the most common patterns of resistance to three and four families in the high-intensity area. Red kite exposure to active antibiotic residues ingested from carcasses of medicated livestock, as occurs in vultures and other scavengers from the same area [15,16,17], could explain these differences between low- and high-intensity farming areas. An ultimate test of this hypothesis would require specific testing for circulating antibiotic residues in red kites, which would confirm their occurrence in the high-intensity area and their lack in the low-intensity areas. In addition, the transference of bacterial resistance from livestock carcasses and other environmental sources, like soils manured with pig slurry frequently sprayed in the high-intensity area and its effect on resistance in wild animals preyed upon or scavenged by red kites, could also contribute to these contrasting resistance patterns. Overall, while resistance selection due to chronic exposure to antibiotics and cross-transmission from livestock may be continuously occurring in the area with factory farms, the ingestion of active antibiotics cannot be a major factor involved in the resistance levels found in the low-intensification area, because no intensive farm or carcass dumps existed in this area. Further research is needed to evaluate the role of chronic low-dose antibiotics ingested as cocktails of active residues on wildlife bacterial resistance.

About twice as many different patterns of resistance to one or two antibiotic families were found in the area with low-intensity farming. This comparatively low multi-resistance can be explained by the lack of important contamination sources with resistant bacteria in agricultural soils and wild animals (especially wild lagomorphs) that constitute the main food source for red kites in this area [36]. The less frequent isolates with resistance to three or more antibiotic families can be attributed to individuals moving from the high-intensity area, which is relatively nearby (about 75 Km, see location map in [32]). Part of the bacterial resistances could be acquired in the breeding areas of the sampled wintering individuals. However, we compared two wintering populations at relatively near geographic areas, thus likely formed by individuals from the same breeding areas in central Europe due to overwintering mixing. Therefore, if resistances were acquired in the breeding areas, we should not expect differences between both populations at the end of the wintering period when the sampling was conducted. This may depend on the residence time of the resistant bacteria, which is generally unknown for wild birds [44]. Other contamination sources of active antibiotic residues and bacterial resistance such as urban rivers [45] could play a role in resistance across the kites’ daily movements. In contrast, the most frequent multi-resistant combinations in the high-intensity area suggest in situ exposure to antibiotics exerting a high selective pressure on bacteria from livestock and red kites. Accordingly, the most frequent multi-resistance patterns found in the high-intensity area resemble those typically found in intensive swine and poultry farms in Spain [46,47], which have been highlighted as amplifiers of antibiotic-resistant pathogens among wild birds [11,48,49]. Further insight on microbial genes encoding resistance coupled with data on its direct and indirect flow from food animals to wildlife can help to fully understand these complex interactions and their consequences in public health and environmental conservation. Specific research on the movement patterns of red kites and other scavengers between areas with different farming schemes can help in our understanding of the spread of bacterial resistance associated with livestock operations.

Resistance in pathogenic microbes to marketed antibiotics began about sixty years ago. Since then, the use of antibiotics in human and veterinary medicine increased in parallel to bacterial resistance [18]. We found that resistance was positively correlated with the time elapsed since the beginning of the commercial application of each antibiotic in human and veterinary medicine, being highest for the oldest antibiotics tested (e.g., neomycin) and lowest for the newest (e.g., imipenem). This time pattern mirrors the historical trend of antibiotic-resistance in human and veterinary medicine [28], although with clear differences depending on farming intensification between areas. A less pronounced slope of the still significant correlation between antibiotic resistance and antibiotic age was found in the low-intensity area than in the high-intensity area. Whether these differences are triggered by the direct exposure to active antibiotics, the cross-transmission of resistance from livestock carcasses and manured soil—or a combination of these processes—requires further research to fully understand the impact of livestock farming on resistance in scavenger wildlife. The strength of these relationships can contribute to calibrating the magnitude of the concern about resistance in farmland wildlife, especially in avian scavengers exposed to variable sources of microbial and pharma-pollution. Specifically, this resistance–time relationship in the high-intensity area can be representative of a strong selective pressure on wildlife Enterobacteriaceae due to anthropogenic inputs into the local environment, especially from factory food animal production.

## 5. Conclusions

Many questions have been raised concerning the impact of the release of antibiotics and resistant bacteria on the environment or on human and animal health [19,50]. Antibiotic resistance in wildlife can be used as an indicator of the impact of human activities on this global health issue [13], especially due to resistance carriage and transmission by birds given their daily long-distance movements and seasonal migrations between continents [22,51]. Because red kites are highly mobile and long-lived, resistant pathogens acquired from intensive livestock operations in wintering areas [49,50] can be host-amplified and spread to distant breeding areas in central Europe, and vice versa, with potential implications in pathogen transmission threatening their declining populations. Indirect effects on health and survival derived from consuming carrion of intensively medicated livestock include the alteration of the normal protective microbiota and the acquisition of antibiotic-resistant bacteria [11,30,31]. Monitoring the consumption of antibiotics, emerging resistances and their dissemination vectors in farmlands is being implemented worldwide given the growing concern of resistance associated with intensive food animal production [3,5]. European programs evaluating these risks are encouraged to use red kites and other avian scavengers (e.g., vultures) as valuable sentinels of contamination with antibiotics and clinically relevant resistant pathogens from livestock operations of variable intensities. In respect to the goal of reducing co-selection of virulence traits, a wise precautionary principle would be to reduce and adequately use antibiotics in prophylaxis and disease treatment in farms authorized to supply carcasses for supplementary feeding of threatened scavengers.

## Figures and Tables

**Figure 1 ijerph-17-03620-f001:**
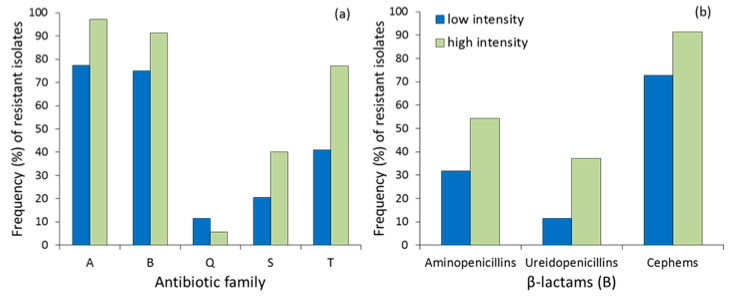
(**a**) Frequency (%) of resistant *E. coli* isolates to each antibiotic family (A= aminoglycosides, B = β-lactams, Q = quinolones, S= sulfonamides, T = tetracyclines) and (**b**) subfamily of β-lactams in red kite droppings from low- and high-intensity farming areas in Central Spain.

**Figure 2 ijerph-17-03620-f002:**
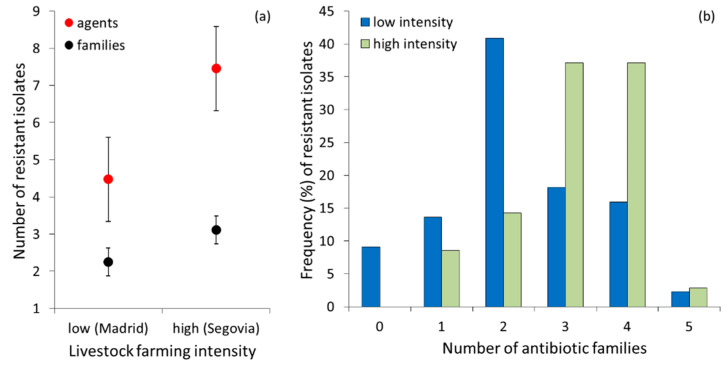
(**a**) Mean number (± SE) of resistant *E. coli* isolates to different agents and antibiotic families in red kite droppings sampled in low- and high-intensity farming areas in Central Spain. (**b**) frequency (%) of resistant isolates to each number of antibiotic families in the low- and high-intensity farming areas.

**Figure 3 ijerph-17-03620-f003:**
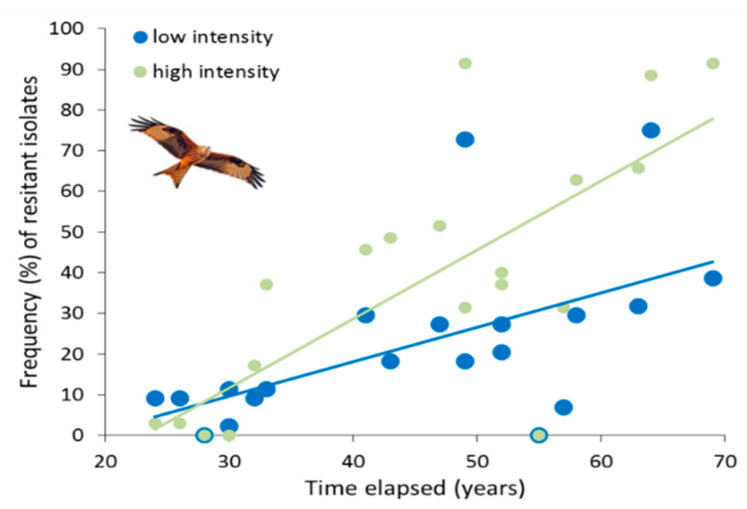
Relationships between the frequency (%) of resistant *E. coli* isolates to each antibiotic in red kites and the time (in year) elapsed since the beginning of the commercial application of each antibiotic. Least squares regression lines of the correlations are shown for graphical representation of trends corresponding to the low- and high-intensity farming areas in Central Spain. Two pairs of overlapping data points with resistance frequency = 0 in both the low- and high-intensity areas correspond to colistin and imipenem.

**Table 1 ijerph-17-03620-t001:** Antibiotic resistance of *E. coli* isolates from red kite droppings sampled in two areas with low (Madrid) or high (Segovia) livestock intensification farming. The numbers of resistant and susceptible isolates between areas were compared with Fisher’s exact tests and relative risk (RR) with 95% confidence intervals (CI). Significant differences are highlighted in bold.

Antibiotics		Livestock Intensification			
Family (Acronym) Subfamily	Agent	Low Madrid (*n* = 44)	High Segovia (*n* = 35)	Fisher Exact Test	
		No. resistant (%)	No. resistant (%)	*p*	RR (95% CI)
**Aminoglycosides (A)**	Gentamycin	8 (18.18)	11 (31.43)	0.194	2.063 (0.724−5.876)
	Kanamycin	3 (6.82)	11 (31.43)	**0.007**	6.264 (1.588−24.710)
	Streptomycin	17 (38.64)	32 (91.43)	**<0.0001**	16.941 (4.481−64.052)
	Neomycin	33 (75.00)	31 (88.57)	0.156	2.583 (0.744−8.971)
**β-lactams (B)**					
Aminopenicillins	Amoxicillin	13 (29.55)	16 (45.71)	0.163	2.008 (0.794−5.081)
	Amoxicillin/clavulanic	4 (9.09)	6 (17.14)	0.325	2.069 (0.535−8.000)
	Ampicillin	12 (27.27)	13 (37.14)	0.466	1.576 (0.607−4.091)
Ureidopenicillins	Piperacillin	5 (11.36)	13 (37.14)	**0.014**	4.609 (1.450−14.648)
Cephalosporins	Cephalothin	32 (72.73)	32 (91.43)	**0.045**	4.000 (1.030−15.534)
	Cephalexin	8 (18.18)	17 (48.57)	**0.007**	4.250 (1.543−11.704)
	Ceftazidime	1 (2.27)	0 (0.00)	1.000	0.551 (0.451−0.673)
Carbapenems	Imipenem	0 (0.00)	0 (0.00)	–	–
**Polypeptides (P)**	Colistin	0 (0.00)	0 (0.00)	–	–
**Quinolones (Q)**	Norfloxacin	5 (11.36)	0 (0.00)	0.063	0.527 (0.425−0.654)
	Ciprofloxacin	4 (9.09)	1 (2.86)	0.376	0.294 (0.031−2.759)
	Enrofloxacin	4 (9.09)	1 (2.86)	0.376	0.294 (0.031−2.759)
**Sulfonamides (S)**	Sulfamethoxazole–trimethoprim	9 (20.45)	14 (40.00)	0.081	2.593 (0.957−7.026)
**Tetracyclines (T)**	Tetracycline	13 (29.55)	22 (62.86)	**0.006**	4.036 (1.571−10.363)
	Oxytetracycline	14 (31.82)	23 (65.71)	**0.003**	4.107 (1.599−10.548)
	Doxycycline	12 (27.27)	18 (51.43)	**0.037**	2.824 (1.105−7.213)

**Table 2 ijerph-17-03620-t002:** Characterization according to resistance patterns by antibiotic family of *E. coli* isolated from feces of red kites from low- and high-intensity farming areas in central Spain.

Livestock Intensification			
Resistance Pattern by Antibiotic Family ^a^	Low (Madrid) No. (%)	High (Segovia) No. (%)	Total No. (%)
Susceptible	4 (9.09)	–	4 (5.06)
A	1 (2.27)	3 (8.57)	4 (5.06)
B	4 (9.09)	–	4 (5.06)
T	1 (2.27)	–	1 (1.27)
A-B	13 (29.55)	5 (14.29)	18 (22.78)
A-Q	1 (2.27)	–	1 (1.27)
A-S	1 (2.27)	–	1 (1.27)
A-T	2 (4.55)	–	2 (2.53)
B-T	1 (2.27)	–	1 (1.27)
A-B-Q	1 (2.27)	–	1 (1.27)
A-B-S	1 (2.27)	–	1 (1.27)
A-B-T	6 (13.64)	12 (34.29)	18 (22.78)
B-S-T	–	1 (2.86)	1 (1.27)
A-B-Q-T	1 (2.27)	1 (2.86)	2 (2.53)
A-B-S-T	5 (11.36)	12 (34.29)	17 (21.52)
A-Q-S-T	1 (2.27)	–	1 (1.27)
A-B-Q-S-T	1 (2.27)	1 (2.86)	2 (2.53)
Total	44 (100)	35 (100)	79 (100)

^a^ A= aminoglycosides, B = β-lactams, Q = quinolones, S= sulfonamides, T = tetracyclines.

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
