# Peer review of "Avian Scavengers as Bioindicators of Antibiotic Resistance Due to Livestock Farming Intensification"

_ijerph, 2020, doi:10.3390/ijerph17103620_

Round 1

Reviewer 1 Report

In this study, the authors evaluated whether contrasting intensification in livestock management influences antibiotic resistance in the Red Kite in farming practices, and assessed the value of avian scavengers as bioindicators of contamination with antibiotics. The manuscript is generally well-written, and the experiment design and results are also interesting.

Comments:

  1. Red Kit, or red kit, it should be consistent throughout the manuscript.
  2. If the E. coli was isolated, and suffered the antibiotic resistance identification? The authors should clarify it in the “2.3 Microflora culture and antibiotic resistance identification”.
  3. The conclusion section is too long, and it should be more concise.

Author Response

Response to the reviewer coments:

  1. We have changed to Red kite throrought the manuscript.
  2. We have clarified in the section 2.3 that we refere to antibiotic resitance in Escherichia coli, as the reviewer suggested. It was previously noted also in the Intoduction section.
  3. We have shorten the text in the conclusion section, as the reviewer suggested .

Reviewer 2 Report

In Line 174, I can see the caption for figure 1; however I do not see any figure 1 in the manuscript. Please include the appropriate figure 1. I am surprised that author is only talking about figure 1(a), where in figure 1(b)? It is hard to figure out if actually the figure 1 is multi-panels.

Authors are encouraged to write a few sentences about the limitations of this research study in the Discussion Section.

Author Response

Response to the reviewer comments:

We think this could be a problem with the format of the template. The Figure 1 was in the original version of the manuscript and in the new version. Figure 1a is cited in the line 168, and the Figure 1b is cited in the line 173 in the new version of the manuscript.

We have added two sentences in the Discusion section about the limitations of the study, and on forms to approach these limitations, including the study of resistance genes and by tracking the movements of Red kites.

Reviewer 3 Report

The article looks at the effect that different levels of industrial food animal production has on the level of antimicrobial resistance in the scavenging Red Kite bird species.

The paper is an interesting study on the potential of scavenging avian species to spread antimicrobial resistant bacteria in the environment.

Major Comments 

The authors state "Detailed descriptions of the study areas and the red kite populations are provided in previous studies [32,36]" Please give an overview of these in this work.

The authors state that "Segovia area has traditionally been used by large numbers of wintering red kites (about 2300 individuals in the study period) gathering in multiple communal roosts" How can the authors be sure that the higher levels of resistance found in the Red Kites here are due to the intensive farming levels of the area and not due to the birds migrating from other areas?

Are all of the Red Kites found in the Madrid test area "local"?

In the Materials and Methods the authors state "Samples were cultured with standard methods as described previously [39]." This is insufficient, please give a full overview of these methods.

The authors should link the antibiotics listed in Table 1 back to the Materials and Methods (Antibiotics used in this study can be seen in Table 1).

The authors should add a brief explanation as to why these 20 antibiotics were chosen.

The authors used "Performance standards for antimicrobial susceptibility testing: 18th informational supplement. CLSI document M100–S18, Wayne, PA. 2008." to determine resistance of their isolates. These standards have undergone several updates since then. Could the authors ensure that their procedures and results comply with the most update standards,

Did the authors only isolate Escherichia coli? This is not clear from the paper.

Minor Comments 

Make sure all bacterial names are italicized

Line 296 - the environment or on human and animal health [19,5o]. This should be 50 presumably?

Author Response

Response to the reviewer comments:

Reviewer coment:

The authors state "Detailed descriptions of the study areas and the red kite populations are provided in previous studies [32,36]" Please give an overview of these in this work.

Authors response: We have modified the sentence to note that we refer to potential impact of feeding on livestock carcasses in supplementary feeding stations in the study area. We have specified these impacts analyzed in other studies.

The authors state that "Segovia area has traditionally been used by large numbers of wintering red kites (about 2300 individuals in the study period) gathering in multiple communal roosts" How can the authors be sure that the higher levels of resistance found in the Red Kites here are due to the intensive farming levels of the area and not due to the birds migrating from other areas?

Authors' response: We agree with the reviewer that part of the bacterial resistances could be acquired in the breeding areas of the sampled wintering individuals. However, we compared two wintering populations at relatively near geographic areas, thus likely formed by individuals from the same breeding areas in central Europe due to overwintering mixing. Therefore, if resitances were acquired in the breeding areas, we should not expect differences between both populations at the end of the wintering period, when the sampling was conducted. We have added a sentence to explain better this conclusion.

Are all of the Red Kites found in the Madrid test area "local"?

Authors response: No. The sampled populationin Madrid correspond to wintering individuals. We have not specified that Red kites from this population are "local". See the above comment.

In the Materials and Methods the authors state "Samples were cultured with standard methods as described previously [39]." This is insufficient, please give a full overview of these methods.

Authors' response: We have added details on the microbiological culture of E.coli. These methods are standard and they have been explined in detail in the cited reference.

The authors should link the antibiotics listed in Table 1 back to the Materials and Methods (Antibiotics used in this study can be seen in Table 1).

Authors' response: Done.

The authors should add a brief explanation as to why these 20 antibiotics were chosen.

Authors response: We have included several sentences to explain why these antibiotics were chosen. Of course, much more antibiotics could have been included in our study, as well as in each published study on bacterial resitance to antibiotics.

The authors used "Performance standards for antimicrobial susceptibility testing: 18th informational supplement. CLSI document M100–S18, Wayne, PA. 2008." to determine resistance of their isolates. These standards have undergone several updates since then. Could the authors ensure that their procedures and results comply with the most update standards,

Authors' response: We use the CLSI document of 2008 because the samples were analyzed in 2013. Some modifications have been included in new versions of these protoccols, but they do not affect our results, as they mostly refer to new antibiotics introduced in human and veterinary medicine. 

Did the authors only isolate Escherichia coli? This is not clear from the paper.

Authors' response: we identified more bacterial species, as cited in reference 39, but we only evaluated resitance to antibiotics in E.coli. 

Minor Comments 

Make sure all bacterial names are italicized

Authors' response: we have italicized all bacterial names.

Line 296 - the environment or on human and animal health [19,5o]. This should be 50 presumably?

Authors response: corrected